# SWE-MiniSandbox: Container-Free Reinforcement Learning for Building Software Engineering Agents

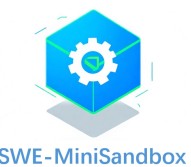

SWE-MiniSandbox

**Danlong Yuan** [1 2 3 4]  **Wei Wu** [3 4]  **Zhengren Wang** [2]  **Xueliang Zhao** [5]  **Huishuai Zhang** [1]  **Dongyan Zhao** [1 6]

Code: https://github.com/lblankl/SWE-MiniSandbox

## Abstract

Reinforcement learning (RL) has become a key paradigm for training software engineering (SWE) agents, but existing pipelines typically rely on per-task containers for isolation. At scale, pre-built container images incur substantial storage overhead, slow environment setup, and require container-management privileges. We propose SWE-MiniSandbox, a lightweight, container-free method that enables scalable RL training of SWE agents without sacrificing isolation. Instead of relying on per-instance containers, SWE-MiniSandbox executes each task in an isolated workspace backed by kernel-level mechanisms, substantially reducing system overhead. It leverages lightweight environment pre-caching techniques to eliminate the need for bulky container images. As a result, our approach lowers disk usage to approximately 5% of that required by container-based pipelines and reduces environment preparation time to about 25% of the container baseline. Empirical results demonstrate that SWE-MiniSandbox achieves evaluation performance comparable to standard container-based pipelines. By removing the dependency on heavy

container infrastructure, SWE-MiniSandbox offers a practical and accessible foundation for scaling RL-based SWE agents, particularly in resource-constrained research environments.

## 1. Introduction

Large language models (LLMs) have demonstrated remarkable capabilities across a broad range of code generation and program synthesis tasks, fundamentally reshaping the landscape of software development and automation in software engineering (Austin et al., 2021; Chen et al., 2021; Guo et al., 2024; Jain et al., 2025a; Li et al., 2022; Liu et al., 2023; 2024; Luo et al., 2025b; Wan et al., 2025). Most existing LLM-based software engineering (SWE) agents adopt container-based execution frameworks to provide isolated and reproducible runtime environments (Xia et al., 2025; Yang et al., 2024a; Wang et al., 2025b; Xia et al., 2024; Jain et al., 2025b). While effective in principle, this paradigm introduces substantial practical overheads: it requires constructing and maintaining a large collection of container images and running high-performance container server clusters, leading to considerable storage, infrastructure, and operational costs. Consequently, scaling to larger batch sizes or higher rollout volumes becomes increasingly costly, with container orchestration emerging as a dominant bottleneck. These limitations hinder scalability under constrained computational resources and exclude users who lack container management privileges or access to dedicated orchestration infrastructure (Luo et al., 2025a; Wei et al., 2025).

To address the scalability and accessibility limitations of container-based SWE evaluation frameworks, we propose a container-free sandboxing system that provides process and filesystem isolation without relying on container or heavy-

[1]Wangxuan Institute of Computer Technology, Peking University [2]Center for Data Science, AAIS, Peking University [3]Ant International [4]Ant Group [5]The University of Hong Kong [6]National Engineering Research Center of New Electronic Publishing Technologies. Correspondence to: Wei Wu <wuwei19850318@gmail.com>, Huishuai Zhang <zhanghuishuai@pku.edu.cn>, Dongyan Zhao <zhaodongyan@pku.edu.cn>.

*Proceedings of the $43^{rd}$ International Conference on Machine Learning*, Seoul, South Korea. PMLR 306, 2026. Copyright 2026 by the author(s).

weight images. Instead of spawning a dedicated container per task, our approach creates an isolated terminal session and a private directory for each instance, enforced via per-instance mount namespaces and `chroot`-based filesystem isolation. On top of this sandbox abstraction, we design an environment pre-caching pipeline that builds lightweight Python venv-based environments, installs task-specific dependencies, and reuses compressed cache artifacts across runs. We carefully manage I/O bottlenecks by packaging environments and repositories into tarball caches, throttling concurrent decompression with Ray-based resource control and semaphores, and supporting multi-node execution to reduce contention. By integrating directly with core SWE tools—SWE-Rex (terminal management), SWE-agent (Yang et al., 2024b) (task solving), and SkyRL (Cao et al., 2025) (scalable multi-node RL)—MiniSandbox functions as a seamless, drop-in replacement for container backends. This design drastically reduces storage usage to about 5% of that required by comparable container-based approaches. Our method also shortens environment preparation time to 25% of container baseline and removes the need for additional container server machines.

Empirically, we show that our framework achieves training performance comparable to state-of-the-art container-based systems, demonstrating strong isolation, good scalability, and accessibility. More broadly, our approach enables a hybrid spectrum of isolation strategies: tasks with stringent system-level requirements can still rely on containers, whereas tasks that can be safely separated using lightweight kernel-based techniques can be executed in virtual-environment–based sandboxes (e.g., via venv or Conda), further reducing resource overhead.

In summary, this paper makes the following contributions:

- **Container-free sandbox for SWE agents.** We introduce a lightweight sandbox using mount namespaces and `chroot` for process and filesystem isolation, avoiding container while remaining compatible with existing SWE tooling.

- **Efficient environment caching and I/O.** We design a venv-based preparation and caching pipeline that reuses compressed artifacts, controls I/O parallelism, and scales to multi-node settings, reducing storage and setup overhead.

- **Resource savings with maintained performance.** Our approach uses ∼5% of the storage and ∼25% of the environment preparation time of container-based methods, without degrading training effectiveness or evaluation fidelity.

- **Flexible isolation strategy.** We show that many SWE tasks can run safely in lightweight virtual-environment

sandboxes, reserving containers only for tasks with strict system-level requirements.

## 2. Related Work

### 2.1. SWE Agent Framework

Following the introduction of SWE-bench, SWE-agent (Yang et al., 2024b) was released as an agent framework that provides a complete interaction pipeline for solving software engineering tasks. It is built on top of SWE-Rex (Yang et al., 2024b), a remote execution framework that maintains terminal sessions on local machines or container backends (e.g., Docker, Podman, etc.).

In addition, SWE-agent has been integrated into the SkyRL (Cao et al., 2025) framework to enable reinforcement learning (RL) training. Moreover, a variety of new agent frameworks (Xia et al., 2025; Yang et al., 2024a; Wang et al., 2025b; Jain et al., 2025b; Xia et al., 2024; Xie et al., 2025), and RL recipes (Luo et al., 2025a; He et al., 2025; Da et al., 2025; Zeng et al., 2025a; Hu et al., 2024) have since been proposed.

Although these frameworks can be deployed locally, they typically rely heavily on container-based environments to support batched agent interactions, environment isolation and execution-based rewards. This design consumes substantial container server resources and poses a barrier to users without access to such infrastructure (Luo et al., 2025a; Wei et al., 2025). Furthermore, maintaining per-instance image caches leads to increased memory usage (Pan et al., 2025). While some methods explore execution-free feedback (Wei et al., 2025; Shum et al., 2025; team et al., 2025; Antoniades et al., 2025), executable environments remain essential and cannot be ignored.

### 2.2. Efforts to Scale SWE Environments

SWE-Gym (Pan et al., 2025) manually configures dependencies per task using task-specific configuration files, yielding ∼2.4k task instances with a one-to-one task–image mapping and about 6 TB of storage.

SWE-smith (Yang et al., 2025) automatically converts GitHub repositories into SWE-style tasks by using LLMs to synthesize bugs. By allowing multiple tasks to share a base image, it produces 50k tasks with only ∼295 GB of images.

SWE-Mirror (Wang et al., 2025a) further improves image reuse via Task mirroring, decoupling tasks from individual images and yielding 60k tasks with ∼100 GB of images. Other works also propose efficient SWE task and environment construction (Zeng et al., 2025b; Guo et al., 2025; Zhu et al., 2025; Hu et al., 2025; Badertdinov et al., 2025).

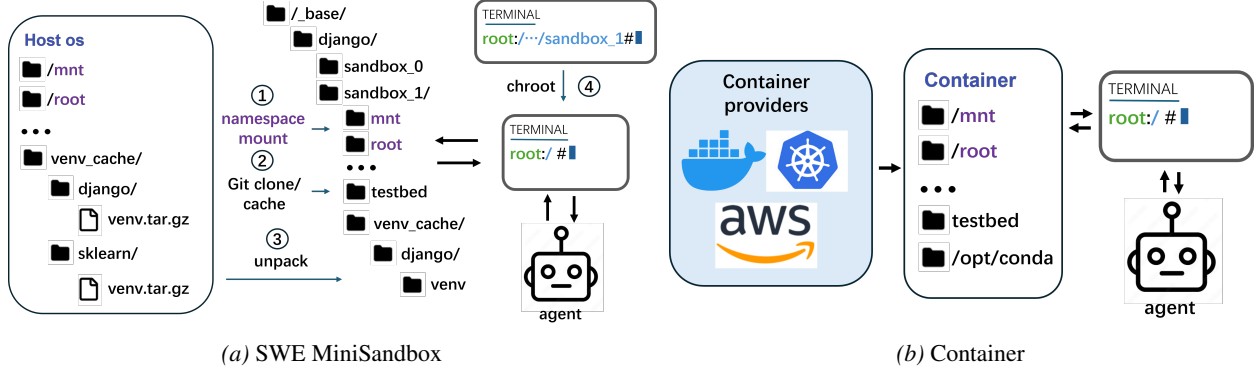

*(a)* SWE MiniSandbox          *(b)* Container

*Figure 1.* Agent Isolation Strategies: Contrasting our per-instance, namespace-based MiniSandbox (left) with conventional container-based isolation (right).

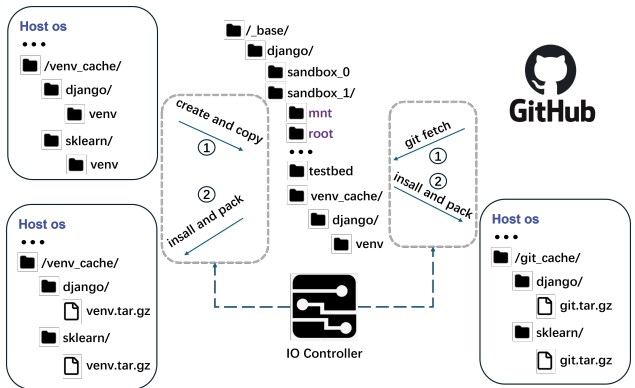

*Figure 2.* Environment Pre-Caching Pipeline: The workflow for building and archiving reusable task environments.

Unlike these image-centric approaches, we focus on the executable environment itself. Observing that most GitHub projects (especially Python) do not require heavy system-level customization, and that lightweight virtual environments (e.g., venv) usually suffice, we propose a container-free framework that uses kernel-level isolation while keeping environment caching lightweight and storage-efficient.

## 3. Preliminaries

SWE-bench (Jimenez et al., 2024) serves as the primary testbed in this study. The benchmark includes verifiable issue-resolution tasks that evaluate the software engineering capabilities of language models.

The data consists of two key components. **Task Context**: Each task includes a GitHub issue (and possibly a related pull request), a snapshot of the repository, and reference patches as ground truth. **Gym Environment**: An executable environment with the target project and dependencies, specified test commands, and evaluation scripts to run tests, verify patches, and compute scores.

The full benchmark spans multiple languages (e.g., Python, C, C++). **SWE-bench Verified** is a curated, Python-only subset of 500 manually verified tasks from 12 repositories, which we adopt for its higher-quality annotations and more reliable evaluation.

A typical SWE-bench pipeline has two stages: (1) *Patch Generation*, where an agent (e.g., an LLM-based coding assistant) interacts with the environment to propose fixes; and (2) *Patch Evaluation*, where the patch is applied to the codebase and the official scripts run the tests and compute the SWE-bench score.

## 4. Method

We introduce SWE-MiniSandbox, a lightweight, container-free sandboxing framework designed for reinforcement-learning–based training of software engineering (SWE) agents. At its core, SWE-MiniSandbox leverages per-instance mount namespaces and chroot to isolate task directories, enabling each task to execute in an independent terminal session without containerization. The framework further incorporates an automatic pre-caching pipeline to maximize environment reuse, mitigates I/O bottlenecks via bounded concurrent decompression, and seamlessly integrates with existing SWE training ecosystems—including SWE-Rex, SWE-agent, and SkyRL—to support efficient and distributed RL training.

Collectively, these design choices eliminate the reliance on large container images that dominate current SWE agent training pipelines. In practice, SWE-MiniSandbox typically requires only ∼100 MB of storage per environment. Its lightweight architecture substantially accelerates environment initialization, its distributed design ensures scalability, and its container-free, Python-only implementation lowers the barrier to adoption, making large-scale SWE agent training more accessible and resource-efficient.

In the following sections, we detail the implementation of

each design choice.

## 4.1. MiniSandbox Launch and Isolation

Unlike traditional container-based approaches that create a separate container for each task (cf. Figure 1b), our method (Figure 1a) establishes an individual terminal session for each sandboxed environment and assigns a dedicated private directory to each task instance.

File-system isolation in our MiniSandbox is achieved using the Linux chroot command which changes the root directory to a specified path, making files outside that directory inaccessible to processes inside the sandbox.

Before invoking chroot, we first create an separate namespace for each agent instance and bind-mount the necessary system directories (e.g., /root, /mnt, /dev) into its private directory. We then copy other required resources—such as the target GitHub project and the corresponding virtual environment—into each private directory. This per-instance mount namespace combined with chroot provides strong isolation while avoiding container overhead.

## 4.2. MiniSandbox Pre-Caching

### 4.2.1. PRE-CACHING PIPELINE

Similar to the image preparation process in container-based SWE frameworks, our MiniSandbox system requires an environment construction stage to be performed in advance. As shown in Figure 2, the main steps are: creating a Python virtual environment (venv), fetching the target Git repository, installing the required dependencies, and then packing the prepared venv back into the cache.

We first create a set of shared miniconda3 installations of different Python versions under a local directory. For each SWE instance, we parse the required Python version and create a corresponding venv-based virtual environment in a designated directory outside the sandbox. This virtual environment is copied into the sandbox directory, while preserving the exact directory structure. This is necessary because many internal paths in a Python venv are hard-coded and cannot be changed without breaking the environment. After that, we install the environment in the sandbox before copying it back to the cache directory.

Since the Python binaries inside the venv are symbolic links to the shared Conda installation, we also bind-mount the shared miniconda3 directory into each sandbox, ensuring that all links remain valid.

We deliberately use Python venv for environment isolation rather than Conda environments. Conda environments are significantly larger and more complex, which would introduce substantial I/O overhead during environment creation

and copying. In contrast, a typical venv for our tasks requires only about 100 MB, meaning the virtual environment is the main persistent artifact we need to store and reuse.

### 4.2.2. I/O BOTTLENECK

After pre-caching, MiniSandbox creation becomes predominantly I/O-bound, with the main costs arising from copying between the virtual environment and the GitHub project directory. To mitigate this overhead, we pre-package these directories into tar.gz archives and reuse them across runs, reducing repeated filesystem operations. However, as the degree of parallelism increases, concurrent decompression itself can become a bottleneck. To mitigate this, we introduce a bounded window mechanism that combines Ray resource tags with thread semaphores, jointly limiting the number of environments that can be unpacked in parallel based on the available I/O capacity.

**Per-task I/O budget model.** We model the disk as having a fixed effective I/O bandwidth $B$ (set to 2000 MB/s in our experiments). For $C$ concurrent decompression tasks, let $b_j$ denote the average I/O throughput of task $j$. To prevent disk saturation, the aggregate I/O demand must satisfy:

$$\sum_{j=1}^{C} b_j \leq B \,. \tag{1}$$

This constraint implicitly determines the maximum admissible concurrency $C^\star$, defined as the largest integer $C$ satisfying Eq. (1).

### 4.2.3. CACHING STAGE OF SWE-BENCH AND SWE-SMITH

An environment is considered successfully prepared only if it passes all provided test cases when evaluated with the golden answer patch. The environment preparation details for SWE-bench are given in Section 6.3.

For SWE-smith, we align our pipeline with its image-reuse strategy. We first identify instances that use unique images (one instance per image) and run our environment cache pipeline on them to cache the base Python venv and the corresponding Git repository. We then process the full SWE-smith dataset: the cached venv is reused, while project repositories are freshly installed and cached as new commits whenever needed.

We filter out instances that already pass (approximately 20k) under the default installation commands. The remaining instances are dropped and not used in subsequent training. This filtering does not imply that the excluded instances are unsolvable; rather, they simply require additional setup or verification beyond the default commands.

*Table 1.* Storage consumption across methods.

| Method | Dataset | #Tasks | #Repos | Env. Storage | Storage Per Task |
|---|---|---|---|---|---|
| One2one | SWE-Gym | 2.4k | 11 | 6 TB | 2.50 GB |
| Task Mirroring | SWE-Mirror | 60k | 40 | 100 GB | 1.60 MB |
| Image Reuse | SWE-smith | 50k | 128 | 295 GB | 5.90 MB |
| One2one | SWE-bench Verified | 500 | 12 | 605 GB | 1.21 GB |
| MiniSandbox (ours) | SWE-smith | 50k | 128 | 13.5 GB | 0.27 MB |
| MiniSandbox (ours) | SWE-bench Verified | 500 | 12 | 89 GB | 178 MB |

*Table 2.* Overall performance and rollout efficiency of SWE-MiniSandbox versus a container-based framework on SWE-Bench. **Reward MD** denotes the mean deviation of the reward during RL training, **Env Prepare Time** refers to the average environment setup time in seconds, and **Avg Rollout Time** is the average rollout time in seconds per instance.

| Model | SWE-Bench Verified | Reward MD | Env Prepare Time | Avg Rollout Time |
|---|---|---|---|---|
| 3B-docker | $5.2 \rightarrow 8.6$ | -0.015 | 88.86 | 367.33 |
| 7B-docker | $7.8 \rightarrow 12.4$ | 0.035 | 90.51 | 355.47 |
| SWE-Agent-7B-docker | $13.4 \rightarrow 16.4$ | 0.016 | 90.12 | 342.41 |
| 3B-MiniSandbox | $5.8 \rightarrow 9.2$ | 0.015 | 23.62 | 272.71 |
| 7B-MiniSandbox | $7.0 \rightarrow 11.8$ | -0.035 | 23.80 | 252.64 |
| SWE-Agent-7B-MiniSandbox | $13.4 \rightarrow 16.8$ | -0.016 | 23.08 | 291.17 |

### 4.3. RL Integration and Distributed Training Behavior

We integrate our MiniSandbox framework into three representative SWE systems: SWE-Rex, SWE-agent (Yang et al., 2024b), and SkyRL(Cao et al., 2025). Terminal management is built on the pexpect-based interaction layer from SWE-Rex, allowing agents to interact with each sandboxed environment through a persistent terminal session.

Agent–RL-environment interaction is implemented in Ray remote functions, enabling MiniSandbox creation and execution to be distributed across all nodes in a multi-node training setup, thus providing high scalability. To control I/O latency, we assign dedicated Ray resource tags and quotas to environment-preparation tasks on each node, thereby limiting the degree of parallel I/O and avoiding filesystem contention. During rollouts, sandboxes are scheduled to utilize available resources across the cluster, without requiring any additional container runtime or orchestration infrastructure such as Kubernetes or managed cloud container services (e.g., AWS ECS/EKS).

## 5. Experiments

### 5.1. Experimental Setup

Our experiments are primarily built upon SWE-agent, SWE-Rex, and Sky-RL. We re-implement the agent interaction pipeline (following SWE-agent), the evaluation pipeline, and the RL training pipeline (following Sky-RL, SWE-bench, and SWE-agent) on top of our MiniSandbox framework.

To evaluate the framework itself, we first focus on the supervised fine-tuning (SFT) stage. We use SWE-agent-LM-32B (Yang et al., 2024b) as a teacher model to generate 5k golden (resolved) trajectories on the SWE-smith dataset, collected independently under both our MiniSandbox framework and the standard container-based framework. Based on these two datasets, we fine-tune Qwen2.5-3B-Coder-Instruct and Qwen2.5-7B-Coder-Instruct (Hui et al., 2024) for 2 epochs, obtaining four student models.

Next, we perform on-policy RL training on each SFT model and the official SWE-Agent-7B (Yang et al., 2024b) model (Qwen2.5-7B-Coder-Instruct finetuned on trajectories from Claude) using 1,600 SWE-smith instances for 1 epoch under both the official container-based framework and our MiniSandbox framework. We set the RL batch size to 16 and the rollout count to 8, resulting in 128 parallel, isolated environment instances per update. All experiments are conducted on a single node equipped with 8×B200 GPUs, 184 CPU cores, and an 800GB SSD. The Docker-based container server used for model serving has 32 CPU cores and a 2TB SSD (the largest configuration available to us), which is sufficient for our workload. For a fair comparison, we cap MiniSandbox's CPU usage at 32 cores in these experiments.

Our RL training adopts a rule-based reward design with the detailed reward signal defined in Appendix A.1. Moreover, we limit the maximum agent interaction time to 300 seconds.

All methods are evaluated using the official container-based SWE-bench Verified pipeline (Yang et al., 2024b; Jimenez et al., 2024). For example, under the "Model 3B-docker" setting, "5.2→8.6" means the SFT version of Qwen2.5-

*Table 3.* Detailed timing breakdown for SWE-MiniSandbox and container-based baselines (in seconds). **Agent Time** denotes average agent interaction time per instance, **Env Time** refers to the average environment communication time per step, **Timeout Times** is the average number of environment communication timeouts per instance, and **Reward Time** means the average reward computation time per instance.

| Model | Agent time | Env Time | Timeout Times | Reward Time |
|---|---|---|---|---|
| 3B-docker | 277.46 | 0.22 | 0.93 | 14.89 |
| 7B-docker | 263.50 | 0.23 | 0.88 | 16.35 |
| SWE-Agent-7B-docker | 251.60 | 0.22 | 0.83 | 16.11 |
| 3B-MiniSandbox | 248.35 | 0.25 | 0.39 | 9.96 |
| 7B-MiniSandbox | 227.90 | 0.25 | 0.31 | 11.32 |
| SWE-Agent-7B-MiniSandbox | 267.08 | 0.21 | 0.34 | 10.12 |

3B-Coder-Instruct achieves a score of 5.2, while the RL-finetuned version reaches 8.6. In addition to accuracy metrics, we report the average environment preparation time in seconds (Env Prepare Time), the average rollout time in seconds during RL training (Avg Rollout Time), and the mean deviation in rewards between a model trained under one framework and its counterpart trained under the other framework over the course of RL training (Reward MD).

Furthermore, we provide a detailed timing breakdown (in seconds) to characterize the performance of our framework, including the average agent interaction time per instance (Agent Time), the average environment communication and execution time per step (Env Time), the average number of environment execution timeouts after 60 seconds per instance (Timeout Times), and the average reward computation time per instance (Reward Time). We also compare storage usage across different methods.

### 5.2. Main Results

Table 1 shows that existing image-based methods, although they reduce memory usage through various optimization techniques, still require substantial storage. In contrast, our MiniSandbox system eliminates the need to store full images, cutting the environment cache size to roughly 5% (13.5 GB vs. 295 GB) and 15% (89 GB vs. 605 GB) of that of image-based approaches on SWE-smith and SWE-Bench Verified, respectively, while remaining compatible with image-reuse strategies such as those employed by SWE-smith (see in Section 4.2.3). For reference, the table also reports the storage usage averaged per instance (Storage Per Task).

As shown in Table 2, our framework achieves evaluation performance comparable to that of the container-based baseline, indicating that both the trajectories generated in our MiniSandbox and the subsequent RL training are of similar quality and effectiveness.

In terms of efficiency, the average environment preparation time in our MiniSandbox is only about 25% of that in the container-based setup (23.62 s vs. 88.86 s).

The detailed timing results in Table 3 further show that reward computation in the MiniSandbox is also faster. At the same time, environment communication time and timeout times are comparable between the two configurations, providing additional evidence that our MiniSandbox framework offers a reliable and efficient execution environment.

In our MiniSandbox, environment initialization involves relatively few kernel-level operations: rather than setting up an isolated container namespace, cgroups, or a full filesystem snapshot, MiniSandbox mostly reuses the host kernel and incurs overhead primarily from filesystem I/O. The main operations are extracting or copying the necessary project files, applying patches, and loading dependencies from the cache. As a result, the end-to-end latency is largely bounded by disk throughput and metadata operations; once the required files are in place, the system is immediately ready for execution.

By contrast, the container-based baseline introduces additional kernel- and runtime-level costs before user code can run. These include creating new namespaces (PID, mount, network, etc.), configuring cgroups, setting up a container filesystem (either from layered images or snapshots), and initializing the container runtime and its entrypoint process. Even when the base images are cached, these steps induce extra system calls, context switches, and filesystem operations beyond the pure I/O needed to materialize the task environment itself. Consequently, the container approach exhibits a higher fixed startup overhead, which leads to a substantially larger average preparation time.

## 6. Discussions

### 6.1. Pressure Test and Multi-node Training

We analyze the efficiency of MiniSandbox across different rollout counts and batch sizes, and further examine its scalability via multi-node experiments, with measurements averaged over the first five steps.

As shown in Table 4, our method is consistently more efficient than the container-based baseline. When scaling to

a rollout count of 16 (i.e., 256 parallel environments on a single node—a substantial workload), I/O and CPU contention prevent fully parallel execution of all rollout workers, leading to increased time for both the MiniSandbox and Container-based setups.

When scaling to 2 nodes, our method maintains high efficiency and exhibits near-linear scalability: for the 256-environment configuration, it achieves environment preparation time comparable to running only 128 parallel environments on a single node, as MiniSandbox workloads are effectively distributed across machines. In contrast, the Container-based approach shows only limited performance gains under the same multi-node configuration due to higher per-environment overhead and constrained resource utilization.

## 6.2. Rollout Latency and Environment Setup Breakdown

We further visualize the rollout stage for a single training step in the 3B-RL setting in Figure 3. Specifically, we report results for step 50; In this figure, the blue bars denote environment preparation time, while the orange bars correspond to agent interaction time. From this comparison, we observe that even under fully parallel execution, environment preparation with containers incurs substantially higher latency than with MiniSandbox, demonstrating the significant efficiency advantages of MiniSandbox over container-based frameworks.

To better characterize the latency introduced by MiniSandbox creation, we also break down and average the components of environment preparation time for the 3B-RL setting. Specifically, the components include: **Init Deployment.** Creating a terminal session and executing the commands required for environment isolation (e.g., namespace, mount, chroot, etc.). **Repo. Copy & Reset.** Fetching or unpacking the target GitHub repository from the local cache and resetting it to the desired commit. **Venv Copy.** Unpacking the pre-built virtual environment from a local directory into the sandbox. **Venv Repo Install.** Reinstalling the repository in editable mode when it is not available from cache. **Other.** Additional overhead, primarily due to CPU scheduling and miscellaneous system operations.

Figure 4 compares the time spent in different components. We observe that the **Repo. Copy & Reset** and **Venv Copy** stages together account for nearly half of the total time, primarily due to high I/O throughput. In addition, a large portion of the overhead arises from system-level operations in **Init Deployment**, which accounts for 35.3% of the total time. This cost is primarily driven by kernel-level work required to set up each sandbox, including creating and configuring isolation primitives (e.g., namespaces, mounts, and chroot), initializing new processes and terminal sessions.

At our level of parallelism, these operations incur substantial scheduler activity and context switching, further amplifying their latency. Consequently, even though **Init Deployment** does not involve heavy user-level computation or large data transfers, the accumulation of these system calls and kernel bookkeeping steps makes it a dominant contributor to overall environment preparation time.

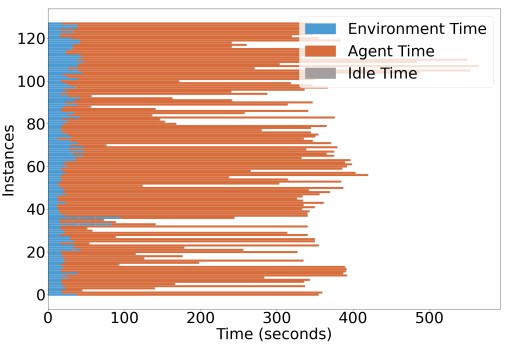

*(a)* SWE-MiniSandbox

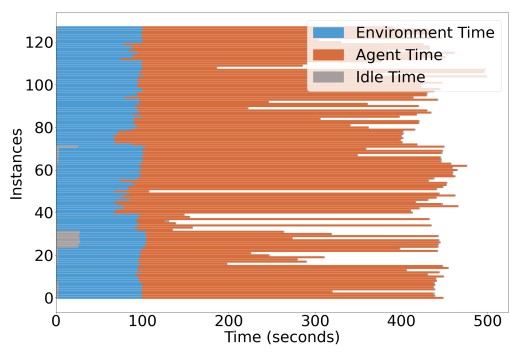

*(b)* Container-based Framework

*Figure 3.* Rollout time comparison between SWE-MiniSandbox and a container-based framework in the 3B-RL setting (step 50).

## 6.3. Evaluation with MiniSandbox

To ensure a fair comparison, the performance reported in Table 2 is evaluated using the official pipeline provided by (Yang et al., 2024b; Jimenez et al., 2024). However, this pipeline relies on a container-based implementation and incurs higher latency than the proposed MiniSandbox framework. This naturally raises an important question: can a comparable evaluation pipeline be built on top of MiniSandbox?

In this section, we investigate this question. Specifically, in evaluation, environment cache in SWE-bench are processed with the following procedure:

1. We first run the default installation commands to verify

*Table 4.* Scalability under increased rollout pressure and multi-node training for SWE-MiniSandbox versus a Container-based framework. **bcs**, **n**, and **Env n** denote batch size, rollout count, and the number of parallel environments, respectively.

| Framework | Nodes | bcs | n | Env n | Env Prepare Time | Rollout Time | Reward Time |
|---|---|---|---|---|---|---|---|
| Docker | 1 | 16 | 4 | 64 | 66.75 | 349.73 | 16.37 |
| Docker | 1 | 16 | 8 | 128 | 87.45 | 361.12 | 16.34 |
| Docker | 1 | 16 | 16 | 256 | 123.79 | 430.80 | 17.15 |
| MiniSandbox | 1 | 16 | 4 | 64 | 11.15 | 227.88 | 10.44 |
| MiniSandbox | 1 | 16 | 8 | 128 | 20.09 | 235.62 | 8.50 |
| MiniSandbox | 1 | 16 | 16 | 256 | 116.25 | 550.80 | 24.17 |
| Docker | 2 | 16 | 16 | 256 | 113.67 | 415.66 | 20.70 |
| Docker | 2 | 32 | 8 | 256 | 117.05 | 415.32 | 18.96 |
| Docker | 2 | 16 | 20 | 320 | 131.92 | 385.23 | 17.84 |
| MiniSandbox | 2 | 16 | 16 | 256 | 20.72 | 259.36 | 16.30 |
| MiniSandbox | 2 | 32 | 8 | 256 | 30.45 | 287.16 | 15.24 |
| MiniSandbox | 2 | 16 | 20 | 320 | 38.73 | 310.41 | 16.20 |

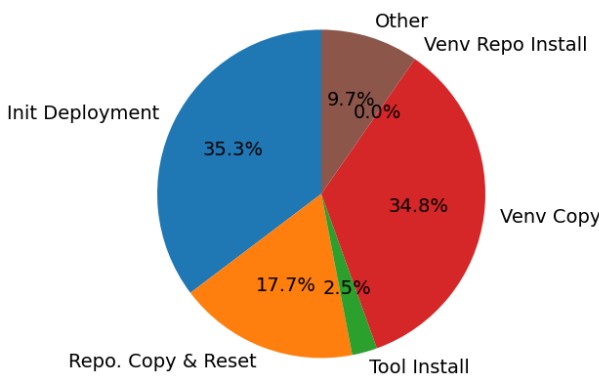

*Figure 4.* Breakdown of environment preparation time components for SWE-MiniSandbox in the 3B-RL setting.

*Table 5.* Evaluation agreement between the container-based (Verify-c) and MiniSandbox (Verify-s) pipelines on SWE-Bench Verified. FN (False Negative): solved by Verify-c only. Exception: disagreements not due to known technical limitations. Note that the official reported results are 15.2 for SWE-Agent-7B and 40.2 for SWE-Agent-32B.

| Model | Verify-c | Verify-s | FN | FP | Exception |
|---|---|---|---|---|---|
| SWE-Agent-7B | 13.4 | 14.8 | 0 | 7 | 0 |
| SWE-Agent-32B | 38.4 | 40.06 | 0 | 11 | 0 |

that each instance can be set up correctly.

2. When necessary, we manually adjust these commands to ensure the environment can be properly configured.

3. Instances that still fail due to network limitations (2 repositories) or hard-to-resolve environment issues are partially excluded by removing the affected pytest cases from evaluation (6 individual instances).

4. A small number of instances fail on a large number of test cases due to difficult-to-diagnose problems; for these 9 instances, we ignore the failures and treat them as passed. Some of these may be fixable with additional engineering effort, but given limited human resources we mark them as system-related. Detailed information is provided in the Appendix A.2.1.

In the worst case, these special instances contribute at most a 3-point fluctuation in the reported metrics.

To further validate our MiniSandbox environment, we select two official models (SWE-Agent-7B and SWE-Agent-32B) and first run inference using the standard container-based SWE-agent framework to obtain their predicted patches. We then evaluate these same patches using both the official container-based SWE-bench framework and our MiniSandbox-based framework. The results are summarized in Table 5.

In the table, **Verify-c** denotes scores obtained with the container-based framework, while **Verify-s** denotes scores from our MiniSandbox framework. The number of instances marked as solved by the container framework but failed by the MiniSandbox is reported as **FN** (False Negative). **Exception** counts how many of these False Negative and False Positive cases cannot be explained by the previously identified system or network limitations.

Overall, the results show that our MiniSandbox framework is consistent with the container-based evaluation, and we observe no unexplained discrepancies, further validating the effectiveness of our framework.

### 6.4. Rationale Behind Environment Isolation and Packaging

**Environment Management with venv:** While many software engineering (SWE) environments rely on Conda for package management due to its robust isolation capabilities, this approach incurs substantial disk overhead by duplicating core components such as the Python interpreter and system libraries. In contrast, Python's built-in venv module offers a lightweight and faster alternative by sharing the base interpreter and isolating only project-specific dependencies. Consequently, a typical venv environment occupies approximately 100 MB, significantly reducing both storage footprint and provisioning latency.

**Compressed Environment Packaging:** To further optimize storage and I/O efficiency, we package environments as compressed tar.gz archives. This strategy minimizes disk usage and network bandwidth during environment instantiation, albeit at the cost of CPU overhead for decompression. The Gzip compression level (1–9) serves as a key tunable parameter for balancing this trade-off: lower levels (e.g., 1) prioritize extraction speed, while higher levels (e.g., 9) maximize space savings at the expense of additional CPU cycles. In our experiments, we adopt the default compression level (6) as a practical compromise between initialization speed and storage efficiency.

### 6.5. High-Concurrency Scaling Behavior

At 256 parallel environments per node (Table 4), the workload exceeds the capacity for strict parallel execution on a single machine, leading to resource contention and increased latency for both setups. Notably, our venv-based MiniSandbox leverages host-level resources for environment startup, whereas the Docker baseline benefits from a dedicated container daemon that can offload certain scheduling tasks; this architectural difference explains the modest performance advantage of containers under extreme concurrency. In multi-node deployments, environment instantiation does not scale linearly due to distributed coordination overhead, including network communication latency, shared storage contention, and cross-node synchronization. Consequently, doubling the parallelism across two nodes incurs a slight efficiency loss compared to the single-node baseline.

## 7. Limitations

Our system is designed primarily for software engineering tasks that operate in user space, so its applicability has several boundaries. First, it relies on Linux-specific mechanisms (e.g., namespaces, mount), so cross-platform support (e.g., Windows, macOS) is currently limited. Second, the isolation boundary is lighter than full containers or VMs, making it less suitable for running untrusted or potentially malicious code, or for tasks requiring kernel-level operations (e.g., device drivers, system module modifications). Third, network isolation is functional but less flexible than Docker's networking stack, which may limit support for tasks requiring complex multi-host topologies or fine-grained traffic control.

## 8. Conclusion

We introduced MiniSandbox, a container-free sandboxing system for evaluating and training LLM-based software engineering agents. By avoiding per-task containers, MiniSandbox reduces storage and setup overhead while remaining compatible with common SWE toolchains. It also supports scalable multi-node execution and provides flexible isolation: lightweight mechanisms are the default, and containers are reserved only for tasks that genuinely require stronger system-level guarantees.

Overall, MiniSandbox aims to lower the barrier to large-scale SWE-agent experimentation by offering an efficient, accessible, and reproducible alternative to heavyweight container orchestration, benefiting both resource-constrained users and teams operating at scale.

One key future direction is to explore an overlay-based filesystem design to further mitigate remaining I/O bottlenecks and improve throughput under high concurrency.

## Impact Statement

This paper presents work whose goal is to advance the field of machine learning. There are many potential societal consequences of our work, none of which we feel must be specifically highlighted here.

## Acknowledgement

The authors sincerely thank the anonymous reviewers for offering invaluable suggestions. The work was supported by National Natural Science Foundation of China (Grant No. 62576016), Beijing Major Science and Technology Project (Z251100008425004) and Beijing Natural Science Foundation (L253001).

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

# A. Appendix

## A.1. RL Training Details

Given a scalar reward function $R(\cdot)$ that evaluates the validity and quality of a generated code patch, let patch denote the final patch produced in an episode. The reward is defined as:

$$\text{reward} = \begin{cases} R(\text{patch}), & \text{successfully submitted,} \\ 0, & \text{otherwise.} \end{cases} \tag{2}$$

## A.2. SWE-bench MiniSandbox Cache Details

### A.2.1. SPECIAL CASES

We summarize the SWE-bench instances that require special handling in our caching pipeline in Tables 7 and 6. Table 7 lists instances for which only specific `pytest` cases are excluded due to hard-to-resolve environment issues or network limitations. Table 6 lists instances that are entirely skipped for similar reasons.

*Table 6.* SWE-bench instances that are fully skipped due to hard-to-resolve environment issues.

| Instance Id |
| --- |
| django__django-14771 |
| sphinx-doc__sphinx-7985 |
| scikit-learn__scikit-learn-14710 |
| django__django-13837 |
| django__django-14311 |
| sphinx-doc__sphinx-10435 |
| django__django-14792 |
| django__django-13809 |
| astropy__astropy-8872 |

### A.2.2. CASE DETAILS

To further clarify the issues mentioned above, we provide some raw test case outputs and briefly analyze their possible causes.

1. **Network errors** (Fig. 5). The test case `psf__requests-2317` reports a network unreachable error such as `ConnectionError: ('Connection aborted.', gaierror(-3, ...))` because the GPU container we use runs in a restricted network environment. We therefore exclude such cases. We also observed other network-related errors, which we resolved by deploying a local `httpbin.core` service. Note that our Docker server does not exhibit these issues.

2. **Environment issues** (Fig. 6). The test case `pydata__xarray-4687` reports an assertion error: `test_duck_array_ops - AssertionError: assert <class 'pint.registry.Quantity'> == <class 'pint.Quantity'>`, which is likely caused by differences in the installed `pint` package version. We did not attempt to resolve this due to limited effort and because it is orthogonal to our main focus.

3. **Hard-to-resolve issues** (Fig. 7). Some problems could not be easily resolved. For example, `astropy__astropy-8872` encounters a deprecation error related to the `pytest` package, and `sphinx-doc__sphinx-10435` reports an `AssertionError`. These failures appear to be tied to upstream package or testing framework changes rather than our environment.

## A.3. Case Study

To illustrate the differences in agent interaction between a normal container and our MiniSandbox, we present several cases that compare the environment responses under the same agent actions.

1. **File editor behavior.** As shown in Fig. 9, we select the same file-editing action (though the internal reasoning traces differ) and extract the corresponding observations from both the Docker framework and our MiniSandbox framework. For clarity, we show only 7 lines and omit the rest. The observations are identical, indicating consistent file editing behavior.

*Table 7.* SWE-bench instances for which specific `pytest` cases are excluded due to hard-to-resolve environment issues or network limitations.

| Instance Id / Repo | Excluded Cases | Reason |
|---|---|---|
| scikit-learn__scikit-learn-14983 | `test_shufflesplit_errors[None-train_size3]` | hard-to-resolve |
| astropy__astropy-7606 | `test_compose_roundtrip[]` | hard-to-resolve |
| pydata__xarray-6992 pydata__xarray-4695 pydata__xarray-3305 | `test_to_and_from_cdms2_classic` `test_to_and_from_cdms2_ugrid` `test_da_name_from_cube` `test_da_coord_name_from_cube` `test_prevent_duplicate_coord_names` `test_fallback_to_iris_AuxCoord` | hard-to-resolve |
| pydata__xarray-4687 | `test_duck_array_ops` | hard-to-resolve |
| psf/requests | `test_mixed_case_scheme_acceptable` `test_conflicting_post_params` `test_pyopenssl_redirect` `test_auth_is_stripped_on_redirect_off_host` `test_mixed_case_scheme_acceptable` `test_requests_history_is_saved` `test_stream_timeout` | network limitations |
| sphinx-doc/sphinx | `test_pyfunction_signature_full_py38` `test_build_linkcheck.py::test_anchors_ignored` `test_build_linkcheck.py::test_defaults_json` `test_build_linkcheck.py::test_defaults` `test_directive_code.py::test_literal_include_linenos` `test_directive_code.py::test_linenothreshold` | network limitations |

2. **Python tool behavior.** As shown in Fig. 10, we select a Python tool invocation and compare its outputs in both environments. Again, we observe no differences, suggesting that Python execution behaves consistently between Docker and MiniSandbox.

3. **`pytest` behavior.** The most notable difference between the Docker environment and our sandbox environment lies in the Python path configuration, which is visible in the `pytest` outputs (see the highlighted orange lines in Fig. 11). Aside from this path difference, the test execution behavior remains aligned across the two environments.

PASSED test_requests.py::test_prepared_request_complete_copy

PASSED test_requests.py::test_prepare_unicode_url

FAILED test_requests.py::RequestsTestCase::test_auth_is_stripped_on_redirect_off_host - requests.exceptions.ConnectionError: ('Connection aborted.', gaierror(-3, '...

FAILED test_requests.py::RequestsTestCase::test_conflicting_post_params - TypeError: 'requests.post(url, data=\\'[{\"some\": \"data\"}]\\', files={\\'some\\'...

FAILED test_requests.py::RequestsTestCase::test_mixed_case_scheme_acceptable - requests.exceptions.SSLError: [SSL: WRONG_VERSION_NUMBER] wrong version num...

FAILED test_requests.py::RequestsTestCase::test_pyopenssl_redirect - requests.exceptions.ConnectionError: ('Connection aborted.', gaierror(-3, '...

FAILED test_requests.py::RequestsTestCase::test_requests_history_is_saved - requests.exceptions.ConnectionError: ('Connection aborted.', gaierror(-3, '...

FAILED test_requests.py::TestTimeout::test_stream_timeout - requests.exceptions.ConnectionError: ('Connection aborted.', gaierror(-3, '...

================== 6 failed, 137 passed, 3 warnings in 1.74s ==================

+ : '>>>>> End Test Output'

+ git checkout 091991be0da19de9108dbe5e3752917fea3d7fdc test_requests.py\nUpdated 1 path from 980e0062

*Figure 5.* The test output from case psf__requests-2317

XPASS xarray/tests/test_units.py::TestDataset::test_computation_objects[int64-coords-method_rolling_exp] numbagg functions are not supported by pint

FAILED xarray/tests/test_units.py::TestPintWrappingDask::test_duck_array_ops - AssertionError: assert <class 'pint.registry.Quantity'> == <class 'pint.Qua...

= 1 failed, 1753 passed, 703 skipped, 109 xfailed, 12 xpassed, 365 warnings in 35.93s =

+ : '>>>>> End Test Output'

+ git checkout d3b6aa6d8b997df115a53c001d00222a0f92f63a xarray/tests/test_computation.py xarray/tests/test_units.py

Updated 2 paths from 8b555e27

*Figure 6.* The test output from case pydata__xarray-4687

from matplotlib import _api, _mathtext

/home/swebench/shared_venv/astropy/astropy/3.1/docker.io/swebench/sweb.eval.x86_64.astropy_1776_astropy-8872/venv/lib/python3.9/site-packages/matplotlib/_mathtext.py:45: in <module>\n ParserElement.enablePackrat()\n/home/swebench/shared_venv/astropy/astropy/3.1/docker.io/swebench/sweb.eval.x86_64.astropy_1776_astropy-8872/venv/lib/python3.9/site-packages/pyparsing/util.py:472: in _inner

    warnings.warn(

E   pyparsing.warnings.PyparsingDeprecationWarning: 'enablePackrat' deprecated - use 'enable_packrat'

========================= short test summary info =========================

ERROR astropy/units/tests/test_quantity.py - pyparsing.warnings.PyparsingDeprecationWarning: 'enablePackrat' deprecated ...

ERROR astropy/units/tests/test_quantity.py - pyparsing.warnings.PyparsingDeprecationWarning: 'enablePackrat' deprecated ...

!!!!!!!!!!!!!!!!!!! Interrupted: 2 errors during collection !!!!!!!!!!!!!!!!!!!!

============================= 2 errors in 3.49s =============================

+ : '>>>>> End Test Output'

+ git checkout b750a0e6ee76fb6b8a099a4d16ec51977be46bf6 astropy/units/tests/test_quantity.py

Updated 1 path from 0e9d85da9c

*Figure 7.* The test output from case astropy__astropy-8872

PASSED tests/test_build_latex.py::test_latex_container

FAILED tests/test_build_latex.py::test_build_latex_doc[lualatex-howto] - AssertionError: lualatex exited with return code 1\nFAILED tests/test_build_latex.py::test_build_latex_doc[lualatex-manual] - AssertionError: lualatex exited with return code 1

FAILED tests/test_build_latex.py::test_build_latex_doc[xelatex-howto] - AssertionError: xelatex exited with return code 1

FAILED tests/test_build_latex.py::test_build_latex_doc[xelatex-manual] - AssertionError: xelatex exited with return code 1

FAILED tests/test_build_latex.py::test_latex_images - AssertionError: assert '\\\\sphinxincludegraphics{{python-logo}.png}' in '%% ...

FAILED tests/test_build_latex.py::test_latex_code_role - AssertionError: assert ((('Inline \\\\sphinxcode{\\\\sphinxupquote{%' + '\\n') +...

======================= 6 failed, 77 passed in 13.09s ========================

+ : '>>>>> End Test Output'

+ git checkout f1061c012e214f16fd8790dec3c283d787e3daa8 tests/test_build_latex.py

Updated 1 path from 5f90f2f52

*Figure 8.* The test output from case sphinx-doc__sphinx-10435

Docker:

"thought": "Great! Let's examine the `parser.py` file to understand the `TextFSMValue` class and the problematic `__str__` method:

"action": "str_replace_editor view /testbed/textfsm/parser.py"

"observation":

"<NOTE>This file is too large to display entirely. Showing abbreviated version. Please use `str_replace_editor view` with the `view_range` parameter to show selected lines next.</NOTE>\n    1 #!/usr/bin/env python\n    2 #\n    3 # Copyright 2010 Google Inc. All Rights Reserved.\n    4 #\n    5 # Licensed under the Apache License, Version 2.0 (the \"License\");\n    6 # you may not use this file except in compliance with the License.\n    7 # You may obtain a copy of the License at\n    8 #\n    9 # http://www.apache.org/licenses/LICENSE-2.0\n    10 #\n    11 # Unless required by applicable

*(a)* SWE-MiniSandbox

MiniSandbox:

"thought":  "Based on the PR description, we need to look at the `parser.py` file since that's where the `TextFSMValue` class is defined. Let's examine this file:"

"action":  "str_replace_editor view /testbed/textfsm/parser.py"

"observation":

"<NOTE>This file is too large to display entirely. Showing abbreviated version. Please use `str_replace_editor view` with the `view_range` parameter to show selected lines next.</NOTE>\n    1 #!/usr/bin/env python\n    2 #\n    3 # Copyright 2010 Google Inc. All Rights Reserved.\n    4 #\n    5 # Licensed under the Apache License, Version 2.0 (the \"License\");\n    6 # you may not use this file except in compliance with the License.\n    7 # You may obtain a copy of the License at\n    8 #\n    9 # http://www.apache.org/licenses/LICENSE-2.0\n    10 #\n    11 # Unless required by applicable

*(b)* Container-based Framework

*Figure 9.* The environment response about file editor tools.

Docker:

"thought": "Let's run the script to see the error:"

"action": "cd /testbed && python reproduce_error.py"

"observation":

\r\n<textfsm.parser.TextFSMValue object at 0x7cc58ba837c0>\n\r\n\r

Sandbox:

"thought": "Let's run the script to see the error:"

"action": "cd /testbed && python reproduce_error.py"

"observation":

\r\n<textfsm.parser.TextFSMValue object at 0x7cc58ba837c0>\n\r\n\r

*Figure 10.* The environment response about python tools.

Docker:

"observation":

============================ test session starts
============================
platform linux -- Python 3.10.16, pytest-8.3.4, pluggy-1.5.0 --
/opt/miniconda3/envs/testbed/bin/python
cachedir: .pytest_cache\nrootdir: /testbed
collecting ...

Sandbox:

"observation":

============================ test session starts
============================
platform linux -- Python 3.10.9, pytest-9.0.2, pluggy-1.6.0 --
/home/smith/shared_venv/google__textfsm.c31b6007/latest/jya
ngballin/swesmith.x86_64.google_1776_textfsm.c31b6007/venv/
bin/python
cachedir: .pytest_cache
rootdir: /testbed
collecting ...

*Figure 11.* Pytest behaivor difference

