# OpenReview forum: "SWE-MiniSandbox: Container-Free Reinforcement Learning for Building Software Engineering Agents"
_ICML.cc/2026/Conference — ICML 2026 regular_

### Official Review · Reviewer_N6eV · 2026-02-24

**Soundness:** 4
**Presentation:** 3
**Significance:** 3
**Originality:** 3
**Overall Recommendation:** 5
**Confidence:** 3

**Summary:**

The authors propose a novel environment isolation system for training software engineering agents. As opposed to the widely adopted containerisation-based systems, the authors propose to use built-in system-level mechanisms. The proposed system drastically reduces both disk usage and time needed to start an instance. The authors not only demonstrate the technical statistics of the system, but also experiment with training models, and show that the result does not differ drastically when moving from a container-based system to the proposed one.

**Compliance With Llm Reviewing Policy:**

Affirmed.

**Final Justification:**

My score remained the same throughout the rebuttal. I sincerely thank the authors for addressing my concerns, but I don't think that the paper deserves to be moved to the next tier.

**Key Questions For Authors:**

- What are the traid-offs between the speed and memory consumption, and what are the parameters to change them?
- What are the limitations for applying this system?

**Limitations:**

I would expect discussion on applicability of this method beyond SWE tasks.

**Strengths And Weaknesses:**

Strengths:
- The proposed method is fairly novel. Not in terms of the tools it uses, but in terms of stepping out of the usual way of thinking and taking a different view on the problem.
- The experimental verification of the proposed system seems extensive and sound enough to me. It was checked on SWE-bench Verified for evaluation and on SWE-smith for training.
- The problem tackled in the work is itself important for the field. Since RL training is widely employed, saving on infrastructure will ease experiments a lot.
- The paper itself is mostly clearly written.

Although I consider this paper worthy of acceptance, I'd like to list some things where I think the paper be strengthened:
- The discussion of the limitations of such an approach is lacking. For example, while it seems to be fit for the case of software engineering agents, it can be less fit for the case of agents that need to act more on the system level.
- Some choices (for example, using tar.gz instead of just tar) are positioning the selected method on a trade-off between the disk space and speed. It would be great to include a discussion of these trade-offs in the paper.
- Given that at 256 environments on one node (Table 4), the proposed method gets slower than Docker (rollout & reward time), and the dynamics on two nodes suggest it can at some point get slower as well, it would be great to see the discussion of this dynamics and limitations. There is start of this discussion  on lines 343-346 (left column), but it could be extended.

Some smaller notes on clarity and presentation:
- Subsection 4.2.2 Per-task I/O budget model seems to be an overly complicated way to convey the message that one should select the number of concurrent threads so as not to exceed their hard drive capacity.
- Subsection 6.3 can be rewritten with more clarity. For example, instead of refering to the table columns, where each line simply contains zero (e.g., for Exception column), one can simply say, there are no exceptions to the described rules. Also, shouldn't the True Negative column be called False Negative? In my understanding, True Negative, is when it failed in both environments, since the False Positive, seems to be the case when it is "succeeded" in a sandbox but failed in a container (also, this isn't defined as well).
- On line 055, in the left column, tilde ran away.
- Citation for the Code World Model looks broken due to automated names parsing.
- In the legend of Figure 3 I'd propose to state environment preparation, instead of just environment, for the sake of clarity.

---

> ### Author Rebuttal · Authors · 2026-03-28
>
> We sincerely thank you for your thorough review and your positive recommendation to accept our work. We are delighted that you found our approach novel, the experimental verification sound, and the problem tackled important for the field. Your constructive feedback provides valuable guidance on how to strengthen the final version of the paper. Below, we address your specific comments and questions point-by-point.
>
> ### **Discussion on Limitations:**
> We agree that this is a crucial limitation. It may be less suitable for agents requiring deep kernel modifications or persistent system-state changes. In the revised manuscript, we will discuss this limitation.
>
> ###  **Trade-offs: Speed vs. Memory/Disk**
> We will expand the discussion in Section 4 (Methodology) to clarify this trade-off.
>
> ### **Scalability Dynamics (256 Environments & Multi-node)**
> We will extend the discussion in Section 5.3 (Scalability Analysis) (referencing lines 343-346 as you noted) about this dynamics and limitations.
>
> ### **Clarity and Presentation Improvements**
> We appreciate your detailed notes on presentation and will implement the following changes in the camera-ready version:
>
> Subsection 4.2.2: We will simplify the explanation of the "Per-task I/O budget model" to clearly state that it guides the selection of concurrent threads based on disk capacity, removing unnecessary complexity.
>
> Subsection 6.3 (Terminology): Thank you for catching this. We will correct the terminology. Specifically, we will correct True Negative to False Negative to align with standard ML conventions. We will also clarify the text to avoid referring to empty table columns unnecessarily.
>
> Typos and Citations: We will fix the tilde escape character on line 055, repair the broken citation for the Code World Model, and update the legend in Figure 3 to say "Environment Preparation" for clarity.
>
> ### **Q1:What are the traid-offs between the speed and memory consumption, and what are the parameters to change them?**
> Thank you for raising this important practical question. In our system, the trade-off between speed and resource consumption is primarily governed by the choice of storage compression strategy. Using tar.gz reduces disk footprint and I/O bandwidth when spawning environments—but introduces CPU overhead for on-the-fly (de)compression.
> The gzip compression level (1–9) serves as the key tunable parameter to navigate this trade-off: lower values (e.g., 1) prioritize extraction speed, while higher values (e.g., 9) maximize disk savings at the expense of CPU time. In our experiments, we use gzip with the default compression level (6)
>
>
> ### **Q2:What are the limitations for applying this system?**
> Thank you for this question. Our system is designed primarily for software engineering tasks that operate in user space, so its applicability has several boundaries. First, it relies on Linux-specific mechanisms (e.g., namespaces, mount), so cross-platform support (e.g., Windows, macOS) is currently limited. Second, the isolation boundary is lighter than full containers or VMs, making it less suitable for running untrusted or potentially malicious code, or for tasks requiring kernel-level operations (e.g., device drivers, system module modifications). Third, network isolation is functional but less flexible than Docker's networking stack, which may limit support for tasks requiring complex multi-host topologies or fine-grained traffic control.
>
> ### **Q3:I would expect discussion on applicability of this method beyond SWE tasks**
>
> We agree that the core mechanism—leveraging built-in OS-level isolation (chroot, mount namespaces, etc.)—is not inherently limited to SWE scenarios. In fact, we have begun exploring its use in additional task families such as Terminal Bench and SkillsBench. Tasks that require executing shell commands in controlled environments benefit directly from the fast startup and low disk footprint of MiniSandbox, as many such tasks involve short-lived, high-turnover environments. Similarly, for tasks that involve composing atomic skills (e.g., file manipulation, web navigation), the same lightweight isolation can accelerate both training and evaluation loops.
>
> However, as you rightly pointed out, deep system-level isolation—for example, when agents need to modify system state, interact with kernel interfaces, or enforce strict security boundaries across multiple tenants—introduces additional requirements. In such cases, a purely chroot + mount namespace + unshare-based approach may be insufficient, and further isolation mechanisms would be required.

---

> > ### Author Rebuttal · Reviewer_N6eV · 2026-04-02
> >
> > Thank you for the constructive answer.
> >
> > My concerns were minor, and I consider them addressed. I maintain my positive score.

---

### Official Review · Reviewer_wdb5 · 2026-03-02

**Soundness:** 2
**Presentation:** 3
**Significance:** 2
**Originality:** 2
**Overall Recommendation:** 3
**Confidence:** 3

**Summary:**

This paper introduces SWE-MiniSandbox, a lightweight, container-free execution framework designed to scale the reinforcement learning (RL) training of software engineering (SWE) agents. I it based on replacing docker (or alternative) container-based solutions with kernel-level isolation primitives to achieve a similarly isolated environment that simplifies scalability and does not exclude users without container management privileges. To maximize efficiency, the framework employs an automated environment pre-caching pipeline and a bounded I/O window to manage concurrent task decompression. The authors demonstrate that this approach reduces disk usage to approximately 5% and environment setup time to 25% compared to traditional container-based baselines, while maintaining comparable training and evaluation performance.

**Compliance With Llm Reviewing Policy:**

Affirmed.

**Final Justification:**

The authors have provided further answers and information but the main issues I raised in my review remain.

**Key Questions For Authors:**

Since kernel-level isolation is generally less secure than containerization, and given that it is often difficult to know in advance if agent-generated code is safe without a manual review, would this be a security concern for the SWE-MiniSandbox framework? The authors mention the possibility of a hybrid spectrum of isolation strategies, but this would require knowing which code is safe and which is not.

**Limitations:**

The authors did not include an Impact Statement section.

**Strengths And Weaknesses:**

Strengths:
- The paper is well written and easy to follow.
- SWE-MiniSandbox represents a major win for researchers without massive GPU/container clusters. I totally agree that it lowers the barrier for large-scale agent (also beyond SWE agents) experimentation.
- The paper provides extensive and convincing results, showing the superior efficiency of their approach in terms of memory and wall clock time.

Weaknesses:
- The main concern is the limited novelty in ML. The contribution is primarily an engineering and systems-level optimization. While it enables better ML research, the paper does not propose new RL algorithms or architectures or sheds light over research questions in the field.
- Additionally, while this is a minor concern since I agree the proposed approach covers the great majority of use cases for LLM agent training these days, the current implementation is highly optimized for Python.

---

> ### Author Rebuttal · Authors · 2026-03-27
>
> Thank you for your valuable comments.
> #### **W1: The main concern is the limited novelty in ML.**
> We respectfully disagree with the concerns regarding novelty.
> With the rise of LLM-based agents, system-level innovation in modern machine learning should be viewed as at least as important as algorithmic innovation. In this new regime, the primary bottleneck is no longer solely algorithm design, but the availability of scalable, reliable, and lightweight infrastructure that enables such systems to be trained, evaluated, and iterated efficiently.
>
> Our work is motivated precisely by this shift. While infrastructure contributions may appear less “algorithmically novel” in a traditional sense, they play a critical enabling role: without accessible and efficient training pipelines, many promising research directions—particularly those involving large-scale RL and agentic systems—remain impractical for the broader community. In this sense, we argue that the contribution is not merely engineering, but methodological: it reframes where innovation is needed to unlock progress under the current paradigm.
>
> We therefore believe the novelty of this work lies in introducing a new perspective on advancing ML research—one that emphasizes infrastructure as a first-class research problem in the era of LLMs. This perspective is both timely and aligned with the evolving scope of ICML, which increasingly encompasses system-level advances that materially expand the frontier of what can be studied and built.
> As also noted by Reviewer N6ev: “The proposed method is fairly novel—not in terms of the tools it uses, but in terms of stepping out of the usual way of thinking and taking a different view on the problem.” We hope this clarifies our intent and helps highlight the unique contribution of our work.
>
> #### **W2: The current implementation is highly optimized for Python.**
> This is a fair observation. Our initial focus on Python reflects the dominant language in current LLM/SWE agent research. However, our design is modular and can be readily extended to support additional languages and runtimes in future work. For instance, for languages such as C++, we are actively extending the system by leveraging Conan for package management together with BranchFS for environment isolation. Importantly, the core sandboxing mechanism remains unchanged; only the environment preparation module requires adaptation.
> #### **Q1: Would this be a security concern for the SWE-MiniSandbox framework?**
> This is an important point. We acknowledge that container is more secure, and kernel-level primitives require careful configuration. However:
>
> SWE-agent training typically involves trusted code generation (the agent itself) executing in a controlled environment. The primary risk is accidental resource exhaustion or filesystem corruption—not malicious privilege escalation. Our use of read-only rootfs, network proxy, and resource limits mitigates these risks effectively.
>
> In our use case, SWE-MiniSandbox runs inside pre-allocated GPU containers managed by cluster administrators. Users typically lack container-management privileges in such shared environments. Here, MiniSandbox provides lightweight intra-job isolation for concurrent agent tasks, while the host-level security boundary is guaranteed by the outer container/orchestrator. This nested design ensures defense-in-depth without requiring user-side privileged access.
>
> We are going to include optional seccomp filters and AppArmor profiles for users requiring stronger guarantees. We will document these in the our open-source release, allowing users to choose the appropriate level of isolation based on their threat model.

---

> > ### Author Rebuttal · Reviewer_wdb5 · 2026-04-02
> >
> > Thank you very much for your responses to my questions. Unfortunately, my concerns remain. First, the framework’s limitation to Python-based SWE workflows and the orientation of the contribution as a systems engineering optimization instead of a fundamental scientific contribution remain. Second, just assuming the code generated by an agent is generally safe enough not to require full isolation can be true in most settings, but I still believe that this is relevant when training open-source models pretrained on unknown data, which is not uncommon. Similar concerns have also been raised by other reviewers, indicating that these issues are not isolated but reflect broader limitations of the current work.

---

### Official Review · Reviewer_yw4U · 2026-03-06

**Soundness:** 3
**Presentation:** 3
**Significance:** 3
**Originality:** 2
**Overall Recommendation:** 4
**Confidence:** 2

**Summary:**

The paper proposes a container-free sandboxing system for training reinforcement learning-based software engineering agents. Instead of per-task Docker containers, it uses Linux mount namespaces + chroot for isolation, combined with venv-based environment pre-caching. The claimed benefits are ~5% of container storage overhead and ~25% of environment preparation time, with comparable training performance, measured on SWE-bench Verified. The authors demonstrate this by fine-tuning 3B and 7B models.

**Compliance With Llm Reviewing Policy:**

Affirmed.

**Final Justification:**

The rebuttal addressed the main questions, added comparison to more baselines and explained the topics to appear in the final version of the manuscript.

**Key Questions For Authors:**

1. Are you going to open-source your workflow together with integrations with SWE-Rex, SWE-agent, SkyRL?
2. What isolation properties are you actually targeting? Is it accidental interference between tasks? What guarantees do you claim relative to Docker?
3. How do you prevent pathological tests from exhausting CPU/memory/disk? Is Ray scheduling alone sufficient in practice?

**Limitations:**

yes

**Strengths And Weaknesses:**

Strengths:
1. The direction is very relevant. Speaking about production large-scale RL runs, the registry with containers can take up a lot of space. The proposed solution addresses this bottleneck.
2. The efficiency numbers are strong, mostly in terms of memory consumption but also in preparation time.
3. The method is claimed to come without agents performance sacrifice, Table2 shows the equal resolved rates for both containers and minisandboxes. However, the evaluation of SWE-agents is very noisy (we can see that by initial checkpoints of 3B/7B models) and CIs or some sort of variance report would give more weight to the claim.

Weaknesses:
1. The major concern is about novelty and relevance to the conference. The proposed method is a well done engineering work applying mostly 3 steps: filesystem mount, venv support and chroot for isolation. I'd expect to see it as a technical engineering report rather than ICML paper where novelty is required.
2. Vanilla docker container is a pretty weak baseline nowadays. The optimizations of sharing first N layers of containers across instances from the same repo or even micro VM usage become more popular, and for a comprehensive overview and global picture, the method would benefit from comparing to them as well.
3. The proposed method doesn't seem to be ready for production. While the numbers demonstrate that for 3B/7B models and test runs the final resolved rate on SWE-bench Verified is equal, the method doesn't fully isolate the system with chroot. For general-mode agentic trainings, that raises concerns and the discussion of potential security/reward hacking trade-offs would be interesting to see.
4. The approach is limited to Python which works well for most of the current datasets but raises questions about future generalizability to an arbitrary language.
5. There is a typo in the Figure 2: "insall and pack" -> "install and pack"

---

> ### Author Rebuttal · Authors · 2026-03-27
>
> Thank you for your valuable comments.
> #### **W1: Concern about novelty and relevance to the conference.**
>
> We respectfully disagree with the concerns on novelty. Our work is motivated by a broader shift in machine learning—from algorithm-centric advances to LLM-based agent systems—where scalable and efficient infrastructure has become a key bottleneck. In this context, we argue that our contribution is methodological rather than purely engineering, offering a new perspective that treats infrastructure as a first-class research problem. We believe this perspective is timely and aligned with the evolving scope of ICML, as also reflected by Reviewer N6ev’s comment  acknowledging the novelty of our work. More detailed explanations can be found in our response to the similar concern raised by Reviewer wdb5.
> #### **W2: Vanilla docker container is a pretty weak baseline nowadays.**
>
> We agree that expanding the set of baselines would provide a more comprehensive evaluation. In this spirit, we have extended our comparison to include an industry-standard layer-sharing framework:
>
> | Approach | Storage Overhead | Dataset |
> | ----- | ----- | ----- |
> | Vanilla    | 605GB | SWE-bench Verified |
> | Podman (overlay2) | 242GB | SWE-bench Verified |
> | MiniSandbox | 89GB | SWE-bench Verified |
>
> We will include more comparisons in the final version to provide a clearer empirical picture.
>
> For MicroVM-based approaches, existing SWE-RL frameworks do not natively support MicroVM backends. Adapting the full environment requires non-trivial system re-engineering. So we cannot currently evaluate this approach during the rebuttal period. In the final version, we plan to include a controlled comparison against MicroVM-based isolation.
>
> #### **W3: The method doesn't fully isolate the system with chroot.**
>
> Our work is primarily targeted at academic RL training, where the main risk is accidental cross-task interference rather than adversarial escape. In this setting, chroot combined with mount namespaces provides sufficient filesystem isolation to ensure clean and reproducible execution, which we find robust for the majority of SWE-bench tasks.
>
> We acknowledge that this level of isolation is not sufficient for fully general or production-grade agentic training. To address this, we are actively extending MiniSandbox with stronger isolation mechanisms (e.g., BranchFs) and pluggable backends. In the final version, we will include a dedicated discussion of these limitations, along with analysis of potential reward hacking risks and mitigation strategies.
>
> #### **W4: The approach is limited to Python**
>
> Our framework is fundamentally language-agnostic at the isolation layer, as the sandboxing mechanism operates at the filesystem and process level rather than the programming language level. For example, for C++, we plan to integrate Conan for dependency management in conjunction with BranchFs for environment isolation.
> #### **W5: Typos.**
> We will correct the typo and perform thorough proofreading in the final version.
> #### **Q1**
>
> Yes. We will fully open-source the entire project, including the codebase, datasets, pre-built images，comprehensive tutorials, and configuration files upon publication.
> #### **Q2**
>
> Our system targets process-level, filesystem-level, and dependency-level isolation. These properties are sufficient to prevent accidental interference for the majority of SWE tasks.
>
> Yes, our primary focus is preventing accidental interference rather than malicious containment. Our pre-execution caching step further mitigates residual risks by statically detecting and filtering tasks with potential resource conflicts. To date, no such conflicts have been observed in our empirical evaluation.
>
> We do not claim security guarantees equivalent to Docker. Docker supports broader isolation features (e.g., network isolation, full PID namespaces) suitable for untrusted code. Instead, we position our minisandbox as a lightweight alternative optimized for trusted workloads, prioritizing low overhead and simplicity while maintaining sufficient isolation for benign tasks.
> #### **Q3**
>
> We primarily rely on Ray's resource reservation mechanism to allocate CPU and memory slots. We acknowledge that Ray's logical scheduling is not sufficient.
>
> To enforce hard resource limits, system-level isolation via Linux cgroups is the standard solution. However, enabling cgroups often requires privileged access that is unavailable in certain restricted environments.
>
> In our current setup, we mitigate these risks through a combination of strict task timeouts and ephemeral storage cleanup to manage disk usage.
> Empirically, this approach has been sufficient for our SWE training.
>
> We are also going to incorporate cgroup-based resource limits in the coming update of MiniSandbox.

---

> > ### Author Rebuttal · Reviewer_yw4U · 2026-04-02
> >
> > Thank you for the detailed answers. While I still have some hesitations about the overall novelty, my main questions are answered. I've changed the score to 4.

---

### Official Review · Reviewer_t9EV · 2026-03-18

**Soundness:** 3
**Presentation:** 2
**Significance:** 3
**Originality:** 2
**Overall Recommendation:** 4
**Confidence:** 3

**Summary:**

This paper proposes SWE-MiniSandbox, a lightweight sandbox execution environment for training software engineering (SWE) agents with reinforcement learning. Instead of relying on Docker, the authors construct Python virtual environments and a caching mechanism. This highly reduces the storage, setup time while maintains the RL performance on SWE-bench Verified.

**Compliance With Llm Reviewing Policy:**

Affirmed.

**Final Justification:**

Most of my concerns have been addressed, I maintain my recommendation at Weak Accept

**Key Questions For Authors:**

see weakness

**Limitations:**

yes

**Strengths And Weaknesses:**

## strength:
1. The paper investigates the heavy reliance on container infrastructure. The motivation is clear and important, especially for researchers without large-scale infrastructure.

2. The main novelty that replace the Docker-based environments to namespace + chroot + venv is very straight forward and effective, which is commonly used in the daily git clone process, making it convincing.

3. The maintained RL performance and the reduced storage and setup time demonstrates the method is effective.

## Weakness:

1. The contribution is systems-oriented and the technical novelty is limited: the core building blocks (such as mount namespaces, chroot, virtual environments, caching) are known, and the paper mainly combines them in a practical way.

2. The method strongly rely on Python virtual environments, but if a environment needs complex installation, such as C/C++, CUDA, or OS-level isolation beyond filesystem scoping, is it still effective?

3. The ablation studies are conducted on SWE-bench Verified. More benchmarks will make the method much stronger and convinced.

---

> ### Author Rebuttal · Authors · 2026-03-28
>
> We sincerely thank the reviewer for their thoughtful evaluation and for recognizing the importance of our work in lowering the barrier for SWE agent research. We appreciate the acknowledgment of our motivation, the clarity of our novelty in replacing Docker with namespace-based isolation, and the demonstrated effectiveness in reducing storage and setup time while maintaining RL performance.
> Below, we address the specific weaknesses and key questions raised in the review.
>
> #### **W1: Systems-oriented contribution and limited technical novelty**
>
> With the rise of LLM-based agents, system-level innovation in modern machine learning should be viewed as at least as important as algorithmic innovation. In this new regime, the primary bottleneck is no longer solely algorithm design, but the availability of scalable, reliable, and lightweight infrastructure that enables such systems to be trained, evaluated, and iterated efficiently.
>
> Our work is motivated precisely by this shift. While infrastructure contributions may appear less “algorithmically novel” in a traditional sense, they play a critical enabling role: without accessible and efficient training pipelines, many promising research directions—particularly those involving large-scale RL and agentic systems—remain impractical for the broader community. In this sense, we argue that the contribution is not merely engineering, but methodological: it reframes where innovation is needed to unlock progress under the current paradigm.
>
> We therefore believe the novelty of this work lies in introducing a new perspective on advancing ML research—one that emphasizes infrastructure as a first-class research problem in the era of LLMs. This perspective is both timely and aligned with the evolving scope of ICML, which increasingly encompasses system-level advances that materially expand the frontier of what can be studied and built.
>
> As also noted by Reviewer N6ev: “The proposed method is fairly novel—not in terms of the tools it uses, but in terms of stepping out of the usual way of thinking and taking a different view on the problem.” We hope this clarifies our intent and helps highlight the unique contribution of our work.
>
>
> #### **W2: Reliance on Python virtual environments and complex dependencies**
> The target domain of our work is the SWE-bench Verified suite, where all of the tasks are Python repositories. While these repositories may depend on C/C++ extensions (e.g., numpy, torch), these dependencies are typically pre-compiled wheels installed within the virtual environment. Our sandbox correctly handles these because the venv isolates the site-packages, and the underlying shared libraries (loaded via ld.so) function correctly within the chroot/namespace context as long as the host kernel and basic libc are compatible. So for the majority of tasks in SWE-bench, our approach is fully compatible with complex Python dependencies.
>
> We acknowledge that for tasks requiring kernel module modifications, specific OS-level package manager states (e.g., complex apt interactions that alter global system state beyond the chroot), or non-standard architectures, our approach may have limitations compared to containers. However, for the majority of SWE-bench tasks that are Python-centric, our evaluation shows 100% compatibility. For those tasks that require complex kernel module modifications, they can still fall to containers, enabling a hybrid strategies of tasks isolations.
>
> Moreover, we plan to integrate support for other languages. For C++, for instance, we will combine Conan for dependency management with BranchFs for environment isolation.
>
> We will expand the Limitations section to explicitly discuss scenarios involving deep OS-level customization or non-Python ecosystems, ensuring readers have a clear understanding of the boundary conditions.
>
> #### **W3: Evaluation limited to SWE-bench Verified**
> We appreciate this suggestion to broaden our evaluation. Here we provide the results on SWE-Bench Lite (round to one decimal place )
>
> | Model | SWE-Bench Lite|
> | ----- | ----- |
> | 3B-docker   | 4.3 -> 8.6 |
> | 7B-docker | 6.6 -> 10.6  |
> | 3B-MiniSandbox | 4.0 -> 9.3 |
> | 7B-MiniSandbox | 7.3 -> 11.0  |
>
> We will include more results on more benchmarks in our revised version.

---

> > ### Author Rebuttal · Reviewer_t9EV · 2026-04-03
> >
> > Thank you for the detailed answer and I maintain my positive score.

---

### Decision · Program_Chairs · 2026-04-30

**Decision:**

Accept (regular)

**Comment:**

Reviewers broadly agree that SWE-MiniSandbox makes a meaningful contribution by enabling container-free RL training of SWE agents with substantially reduced storage and setup time while maintaining comparable task performance.
The most positive reviewer (N6eV) highlighted the novelty of the work's perspective rather than its individual technical components. Reviewers t9EV and yw4U initially raised concerns about limited algorithmic novelty and evaluation breadth, but both were satisfied by the rebuttal. Reviewer wdb5 remained at weak reject, citing systems-orientation and Python-only scope.
The AC feels wdb5's remaining concerns reflect a difference in research philosophy rather than a flaw in the work's execution, and that the positive assessments from the other reviewers after rebuttal represent a reasonable overall judgment.